# Hypertension Detection Based on Photoplethysmography Signal Morphology and Machine Learning Techniques

**Lucian Evdochim** [1,*] , **Dragoș Dobrescu** [1] , **Stela Halichidis** [2] , **Lidia Dobrescu** [1] and **Silviu Stanciu** [3]

1 Department of Electronic Devices, Circuits and Architectures, Faculty of Electronics, Telecommunications and Information Technology, University Politehnica of Bucharest, 060042 București, Romania
2 Department of Clinical Medical Disciplines, Faculty of Medicine, Ovidius University of Constanta, 900527 Constanța, Romania
3 Laboratory of Cardiovascular Noninvasive Investigations, Dr. Carol Davila Central Military Emergency University Hospital, 010242 Bucharest, Romania
* Correspondence: lucian.evdochim@stud.etti.upb.ro; Tel.: +40-722786181

**Featured Application:** **Hypertension detection analyzing the morphology of the PPG signal aided by Machine Learning Techniques.**

**Abstract:** In our modern digitalized world, hypertension detection represents a key feature that enables self-monitoring of cardiovascular parameters, using a wide range of smart devices. Heart rate and blood oxygen saturation rate are some of the most important ones, easily computed by wearable products that are provided by the photoplethysmography (PPG) technique. Therefore, this low-cost technology has opened a new horizon for health monitoring in the last decade. Another important parameter is blood pressure, a major predictor for cardiovascular characterization and health related events. Analyzing only PPG signal morphology and combining the medical observation with machine learning (ML) techniques, this paper develops a hypertension diagnosis tool, named the ANC Test™. During the development process, distinguishable characteristics have been observed among certain waveforms and certain types of patients that leads to an increased confidence level of the algorithm. The test was enchanted by machine learning models to improve blood pressure class detection between systolic normotensive and hypertensive patients. A total of 359 individual recordings were manually selected to build reference signals using open-source available databases. During the development and testing phases, different ML models accuracy of detecting systolic hypertension scored in many cases around 70% with a maximum value of 72.9%. This was resulted from original waveform classification into four main classes with an easy-to-understand nomenclature. An important limitation during the recording processing phase was given by a different PPG acquisition standard among the consulted free available databases.

**Keywords:** hypertension detection; photoplethysmography; machine learning; signal classification; signal database

## 1. Introduction

The present technological trend of digitalization brings multiple advantages across various domains where human activity is developing. One important domain is the medical environment, where a new wide range of smart devices [1] has risen to aim at self-monitoring. This shift in paradigm is given by the increase in popularity of wearable devices which are capable of tracking cardiovascular markers. The most common ones are represented by heart rate (HR) and blood oxygen saturation level (SpO2), followed by blood pressure (BP) evaluation and even an electrocardiogram (ECG) [2,3]. A major contributor to bring such monitoring features closer to user hand is represented by photoplethysmography (PPG) techniques which gains interest and popularity in the last decade across scientific

circles. The obtained electrical signal by illuminating capillaries contains an abundance of cardiovascular information which waits to be interpreted. Except for the previous presented parameters, this technique shows a good confidence level to detect changes in vasomotor and respiratory activities. It was demonstrated that vasodilators or vasomotor drugs modulate the morphology of PPG signals especially by the dicrotic wave behavior [4–6]. Additionally, the respiratory activity plays a major role since it modulates the signal where a connection was found between modulation amplitude and hypertensive patients [7]. Therefore, based on this vasomotor activity, the trend of BP value can be extracted since it is one of the critical health markers [8]. Therefore, multiple studies in this direction started and some of them offer the reference to the used public databases such as China Database, MIMIC III, CapnoBase, and University of Queensland Database [9–12].

Although the search to extract BP prediction from analysing PPG waveform is approached from multiple directions, existing studies do not take into consideration a solid model which connects the mechanical nature of blood pressure with the optical nature of photoplethysmography. Despite those mechanisms being still in debate, especially related to the true origin of PPG and its morphology, it did not represent a blocking point to assess cardiovascular characterization at some confidence level [13–15]. For example, in a cold environment, vasoconstriction on peripheral area is triggered, thus the amplitude of PPG signal decreases [16]. Another factor is detecting certain cardiovascular disorders such as pulsus bisferiens [17] or pulsus alternans which are detected within waveform analysis without knowing exactly the above-depicted models.

In this way, without focusing on the origin topic, we started to define at a first glance, the desired level of cardiovascular information which can be extracted from the optical signal. We identify three levels of prediction, depicted in Figure 1, as follows:

- Level 1—PPG signal can predict BP class (hypotensive, normotensive, hypertensive) in an individual case. This prediction can be scaled-up to a larger population and represent the first assessment in an advanced signal processing topic.
- Level 2—PPG signal can predict BP trend qualitatively (rising or falling) during a certain time frame observation in an individual monitoring case. This level is proven by drug administration during anaesthesia procedure and even by the respiratory cycle where PPG waveform is modulated. This prediction can be also scaled-up to a larger population but involves a large time of signal observation unlike in the first level.
- Level 3—PPG signal can predict BP value. This level predicts the BP trend from a qualitative mean. It is hard to achieve this prediction rate because a change of $+/-15$ mmHg for example, brings up different waveform change in two different persons. Work-around techniques such as PTT which fusions a second sensor like ECG proves a good achievement at this step. This level also requires a calibration step since PPG signal gives to the processing system only optical quantities. The most challenging step is to extract cardiovascular information only from PPG signal standalone with just one calibration procedure during the lifetime of the patient monitoring.

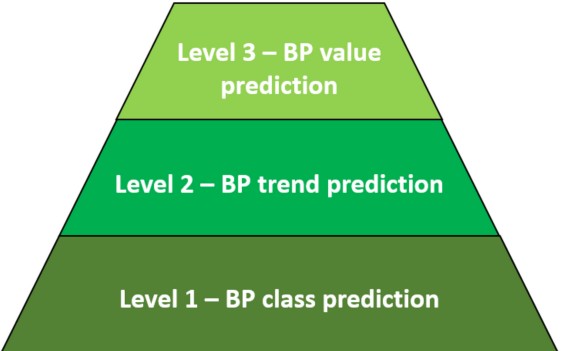

**Figure 1.** PPG performance level for BP assessment.

By climbing the levels, the predicted value range narrows or others say the accuracy increases. Therefore, more advanced signal analysis methods are required to extract this important health metric.

## 2. Methods

### 2.1. Signals Selection

For having a large spectrum of PPG signal morphology, multiple databases should be investigated. The topic addressed by the presented study is to funnel various waveforms into a few main categories sorted by clearly defined criteria. In this way, specific key metrics correlated to certain population group types can be identified. The first step into this investigation was to visualize signals from available free-access databases. We found recordings with different health metrics which are summarized in Table 1. The primary criterion was for datasets to contain PPG signals and associated BP information. Three recording databases have been used to highlight morphology key metrics, but only two where used in the final algorithm development procedure. In total, 359 PPG recordings with corresponding BP class have been used as described:

**Table 1.** Investigated datasets for PPG key metrics extraction.

| Name of Database | Label | No. of Individual Recordings | BP Signal | PPG Signal | Age | Weight | Height |
|---|---|---|---|---|---|---|---|
| China Database [9] | Dataset 1 | 219 | Yes, oscillatory method | Yes, between 3–6 periods | Yes | Yes | Yes |
| MIMIC III [10] | Dataset 2 | 140 | Yes, invasive method | Yes, between 5 and 20 periods | No | No | No |
| CapnoBase [11] | Dataset 3 | 42 | No | Yes, around 100 periods | Yes | Yes | No |
| University of Queensland [12] | N/A | 32 | Yes, intermittent 5 min oscillatory | Yes, around 1000 periods | No | No | No |

Dataset 1 contains a total of 219 recordings from volunteers where the BP distribution is represented in Figure 2a. It is observable that distribution follows a Gaussian trend where normotensive and hypertensive groups are balanced. The protocol of signal acquisition uses a sampling frequency of 1 kHz where 3 clusters of 3 periods were electronically saved. Blood pressure recordings were carried out by oscillometric methods while the patients were sitting in a relaxed position.

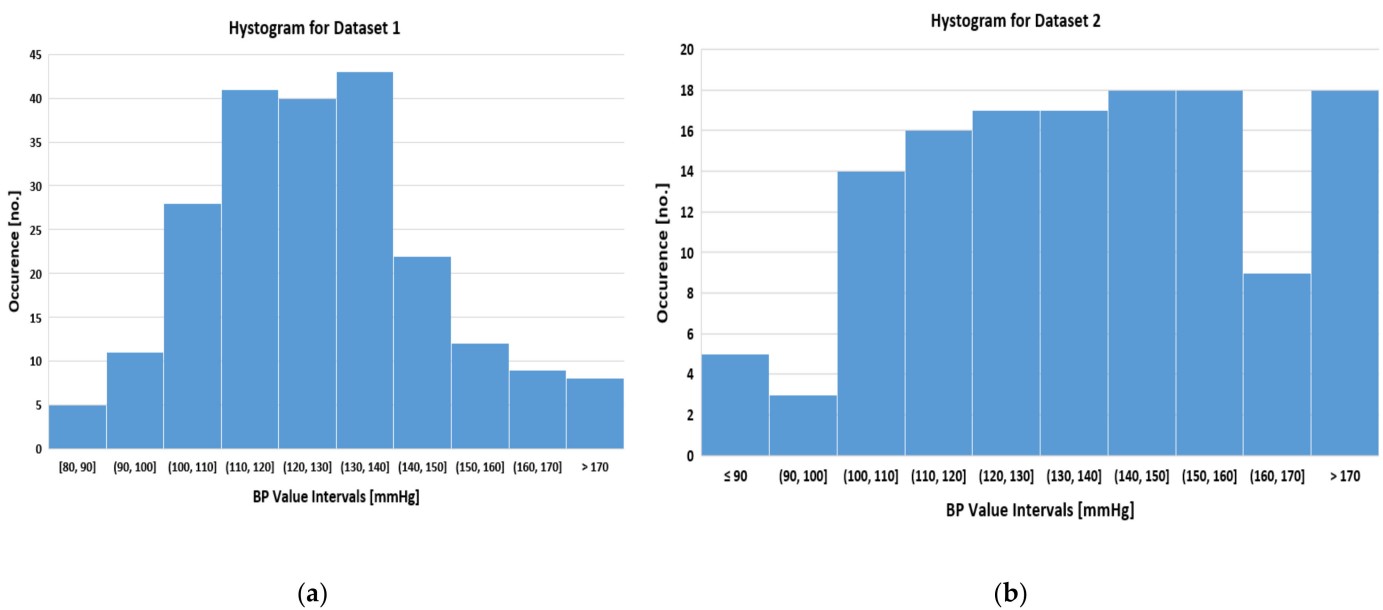

(**a**)  (**b**)

**Figure 2.** Histogram of BP values for Dataset 1 (**a**) and Dataset 2 (**b**).

Dataset 2 contains a total of 12,000 recordings but by an iterative visual inspection it was deduced that clusters of consecutive 10–30 recordings belonged to the same patients. The protocol of signal acquisition uses a sampling frequency of 125 Hz where the periods

vary from 5 to 20 cardiac cycles. Associated BP information was taken by an invasive method using the catheter technique. The monitored patients were in an Intensive Care Unit or in surgery procedure. Therefore, a manual dataset with the same label was created with signals coming from each individual patient. With this procedure, a total number of 140 recordings were electronically sorted where the secondary criterion was to obtain a uniform distribution among BP classes as shown in Figure 2b.

Dataset 3 contains a total number of 42 patients but since the BP information is missing it was not used in the final step of the algorithm development. Its available information was used only to help the presented investigation in the key metrics extraction procedure.

The last dataset which does not have an allocated label was used only for visualizing the PPG morphology under the entire time frame of the surgery procedure. This dataset helps to find the relevant key metrics for the next steps which will be presented.

It is important to note that signals come from motionless context since the patients were being monitored in an Intensive Care Unit or in related stationary cases. However, this topic does not represent a blocking point in healthcare algorithm development. During 24 h a person can certainly experience time frames where motionless events occur, such as sleeping (around 8 h of stationary state), sitting in front of the desk, reading, waiting in front of queue, and other related social activities. Therefore, these scenarios made a major topic for developing an algorithm specialized for PPG signal acquisition. In the presented research the focus is on signal analysis with the assumption that the provided recordings come from stationary events.

### 2.2. Signals Classification

The next step after database selection was to find relevant signal morphology characteristics among different patient groups as was described earlier. The main observations during recordings evaluation in different scenarios, while also having in mind the findings reported in the medical literature [18–22] were:

- Anacrotic limb features are found in many cases in the old patients group or for hypertensive ones. They represent a change in signal slope, PPG, or BP, during the systolic rising phase.
- Dicrotic wave location moves away with respect to SYS points when vasodilators were administrated during the beginning of anesthesia. The vasoconstrictor drugs give the opposite effect by moving the dicrotic wave toward SYS peak. Therefore, this fiducial point can represent the state of vasomotor activity.
- Dicrotic wave is mostly absent in the children group, under 10 years old, and very prominent in young groups with ages between 20 and 30. After these limits, in older populations it fades again.

Another observation, but related to signal acquisition, is the different protocols used for PPG recordings. One important highlight used sampling frequency but also the cutoff frequency for signal filtering. This variation is translated into different cardiovascular details which will be discussed in the Results and Limitations sections. Those differences among databases did not represent a blocking point for the current step but could influence the performance of the final algorithm.

After the signal characteristics arise, we needed a tool to translate it into a numerical value. A good domain to highlight signal morphology was first derivative (FD) since it represents the evolution of the signal gradient over time. This domain has the advantage to show a more detailed level of the investigated waveforms which are not visible by the eye. Therefore, after the initial pre-evaluation steps, we define four major classes among patients without cardiovascular disease which modulated PPG or BP signals such as pulsus bisference. The defined classes take the dicrotic wave and anacrotic feature as the selection criteria:

- Anacrotic type (A)—This class highlights the pre-systolic phase where signal slope is subject to a change [18,19]. This feature is translated into FD domain as a visible peak before SYS maximum point event as it is shown in Figure 3a.

- Normal type (N)—This class highlights the post-systolic phase where DCW generates a visible peak in the original analyzed signal. In FD domain, the associated gradient of the dicrotic wave beginning phase is greater than zero (or positive), building a peak as shown in Figure 3b.

- Collapsing type (C)—Also aims at the post-systolic phase where DCW is entirely absent in the analyzed signal. In FD domain its gradient does not appear as the result of a wave missing. After the SYS event, the returning gradient slowly rises towards zero axis, without generating a peak, until the next cardiac cycle begins [20] as shown in Figure 3c.

- Normal Collapsing (NC) type—This class is a characterization between the previous two. DCW is barely visible in the investigated signal, therefore, it generates a gradient peak into FD domain but negative in absolute value as shown in Figure 3d.

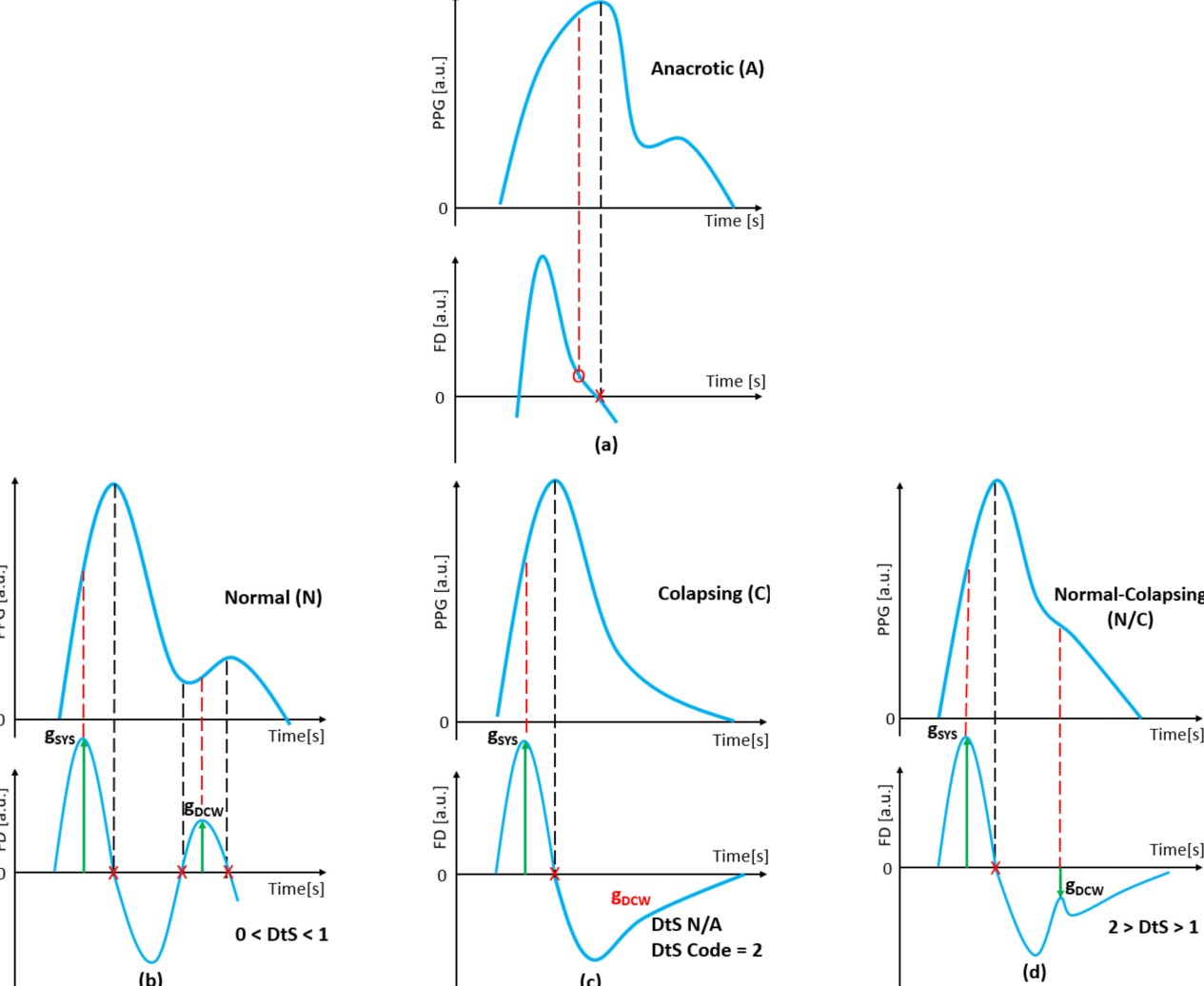

**Figure 3.** PPG signal morphology classification by DCW behaviour in FD domain in healthy patients, also applicable for BP signal. Discussion in the text. Red lines represent the correspondence between original signal and its gradient peak location in FD; the associated amplitude is marked with green arrow; exception for anacrotic case where the gradient is marked with red circle. Black lines represent the correspondence between the original signal and its zero crossing location marked with red cross.

Note that multiple peaks after SYS event can occur due to different elastic characteristic of the arterial network. These peaks can be present even if the signal filtering is applied and are therefore not given by motion or acquisition artifacts. In our classification context,

the dicrotic wave presence is flagged as the strongest gradient in FD domain in the defined search window.

Using this major classification, signals are characterized from a qualitative mean with an intuitive nomenclature. In order to cover a quantitative topic, a numerical term should be associated to gradients since it can range from any positive to negative value. As we anchor classification to the dicrotic wave gradient it needs to be related to another important key metric during the current cardiac cycle. Thus, a numerical parameter was defined, labelled as DtS which stands for Dicrotic to Systolic Ratio. The computed parameter translates into numbers of the human's sense of the observed PPG morphologies during signal evaluation step. In this way, it is characterized how strong the dicrotic wave is with respect to the systolic one according to the presented classifications from Figure 3. The formula for defining this morphological parameter is:

$$\text{DtS} = \frac{g_{\text{SYS}} - g_{\text{DCW}}}{g_{\text{SYS}}}$$

where $g_{\text{SYS}}$ represents the value of systolic gradient extracted from the first derivative and $g_{\text{DCW}}$ is the value of the dicrotic gradient part. While the SYS gradient is always positive, the other ones take both possibilities. This is why the DtS parameters compute the distance between the gradients of the respective fiducial points with respect to the positive ones. With this rule, the translation of defined classes was translated via DtS parameters as presented in Table 2.

**Table 2.** Numerically values associated with defined classifications.

| Class | $g_{\text{SYS}}$ | $g_{\text{DCW}}$ Value Interval | DtS Value Interval |
|-------|------------------|--------------------------------|--------------------|
| N | | $(0:g_{\text{SYS}})$ | $(0:1)$ |
| NC | Always >0 | $(-g_{\text{SYS}}:0]$ | $[1:2)$ |
| C | | N/A | 2 * |

\* It's a convention code since DtS doesn't have this computed value assigned.

Note that DtS is computed per individual cardiac cycle. Thus, in N type signal where $g_{\text{DCW}}$ is defined to be positive, the above parameter will be lower than 1 value. While in NC type were DCW gradient is negative, the parameter will be greater than 1. In other words, DtS highlights how close the dictoric wave gradient is to the systolic one in the FD domain. For C type where dicrotic is entirely absent, DtS cannot be computed but is marked with value 2. This convention comes from the fact that a hypothetically maximum negative value taken by a second wave cannot exceed absolute value of the current systolic gradient. As we described earlier, a strong FD value represents a strong visible point back to the analyzed signal.

If a signal recording is evaluated during multiple periods, the above-described parameters is computed as a mean. With those classifications—nomenclature of waveform morphology by pre-systolic and post-systolic evaluation criteria—we built a test labelled ANC Test™ (**A**nacrotic, **N**ormal, **C**ollapsing) which stands for Level 1 class prediction described back in Figure 1. Hence, the numerical outputs of this designed test, which are intended for multiple cardiac cycles evaluation, are:

- Anacrotic occurrence if at least two cardiac cycles are evaluated within the pre-systolic phase. For a single-period analysis, the algorithm just marks if the Anacrotic feature is present or not.
- Normal, Normal-Collapsing and Collapsing occurrence evaluated in the same manner as above but within the post-systolic phase.
- DtS ratio evaluation for every individual analyzed cardiac cycle. In the collapsing case the ratio is marked with code 2 since mathematically it cannot be computed.

All the occurrence computations are done automatically since one part of the intended algorithm classified recordings by the defined features. In multiple periods of evaluation

mode, the occurrence rate for the presented types can takes any value between 0% and 100%. But in the opposite case where just a single cardiac cycle is analyzed the occurrence rate can take only 0% or 100% value. It is a low probability that a patient's key metrics will be investigated just based on one cardiac cycle, therefore the first mode will be frequently used.

### 2.3. ANC Test Pre-Evaluation

To evaluate what correspondence between defined feature and population group type arises, the built test was run in the following cases: Dataset 1 and Dataset 2 to check the results against BP groups and Dataset 1 and Dataset 3 to check against age groups. The splitting granularity of both groups was chosen to be 10 units. The overall results are presented into Tables 3 and 4 for first group's type and into Tables 5 and 6 for the second one. The test was run in a multiple cardiac cycles evaluation mode where a patient takes the predominant ANC type where the occurrence threshold rate was chosen at 50%. For example, if within a recording length for A type the occurrence was around 60%, the patient would have the respective label. For the post-systolic evaluation, the occurrence sum of N, NC, and C cannot exceed 100%. Therefore, the predominant one was taken as the final label.

**Table 3.** ANC Test results vs. BP groups for Dataset 1.

| BP Group [mmHg] | Total No. | Occ. of A [%] | Occ. of N [%] | Occ. of NC [%] | Occ. of C [%] |
|---|---|---|---|---|---|
| 80–89 | 4 | 25 | 0 | 100 | 0 |
| 90–99 | 7 | 0 | 0 | 100 | 0 |
| 100–109 | 28 | 0 | 14.29 | 82.13 | 3.57 |
| 110–119 | 41 | 4.88 | 7.32 | 90.41 | 2.27 |
| 120–129 | 44 | 13.64 | 2.27 | 95.46 | 2.27 |
| 130–139 | 41 | 14.63 | 2.43 | 97.57 | 0 |
| 140–149 | 24 | 25 | 4.16 | 91.67 | 4.16 |
| 150–159 | 10 | 30 | 9.09 | 90.90 | 0 |
| 160–169 | 11 | 18.18 | 0 | 100 | 0 |
| 170+ | 9 | 33.33 | 0 | 100 | 0 |

**Table 4.** ANC Test results vs. BP groups for Dataset 2.

| BP Group [mmHg] | Total No. | Occ. of A [%] | Occ. of N [%] | Occ. of NC [%] | Occ. of C [%] |
|---|---|---|---|---|---|
| 80–89 | 5 | 80 | 20 | 80 | 0 |
| 90–99 | 3 | 33.33 | 33.33 | 66.66 | 0 |
| 100–109 | 11 | 9.09 | 72.72 | 27.28 | 0 |
| 110–119 | 16 | 0 | 43.75 | 50 | 6.25 |
| 120–129 | 18 | 11.11 | 38.88 | 55.56 | 5.55 |
| 130–139 | 14 | 21.42 | 21.42 | 78.57 | 0 |
| 140–149 | 22 | 31.81 | 9.09 | 85.36 | 4.54 |
| 150–159 | 17 | 17.64 | 23.25 | 76.47 | 0 |
| 160–169 | 11 | 36.36 | 36.36 | 54.54 | 9.09 |
| 170+ | 18 | 44.44 | 33.33 | 61.11 | 5.55 |

**Table 5.** ANC Test results vs. age groups for Dataset 1.

| Age Group | Total No. | Occ. of A [%] | Occ. of N [%] | Occ. of NC [%] | Occ. of C [%] |
|---|---|---|---|---|---|
| 20–29 | 23 | 0 | 26.09 | 73.91 | 0 |
| 30–39 | 5 | 0 | 0 | 100 | 0 |
| 40–49 | 28 | 14.29 | 0 | 92.86 | 7.14 |
| 50–59 | 61 | 13.11 | 3.28 | 95.09 | 1.63 |
| 60–69 | 53 | 13.21 | 0 | 100 | 0 |
| 70–79 | 34 | 17.65 | 0 | 97.05 | 2.94 |
| 80+ | 15 | 40 | 0 | 100 | 0 |

**Table 6.** ANC Test results vs. age groups for Dataset 3.

| Age Group | Total No. | Occ. of A [%] | Occ. of N [%] | Occ. of NC [%] | Occ. of C [%] |
|---|---|---|---|---|---|
| 0–9 | 15 | 0 | 13.33 | 60 | 26.66 |
| 10–19 | 13 | 0 | 76.92 | 23.07 | 0 |
| 30–39 | 4 | 0 | 50 | 50 | 0 |
| 40–49 | 3 | 0 | 75 | 0 | 25 |
| 50+ | 6 | 0 | 0 | 83.33 | 16.66 |

It can be observed from Tables 3 and 4 that A type occurs more frequently in prehyper and hypertensive class (130+ mmHg) with 75.36% probability in Dataset 1 and with 53.18% in Dataset 2. For N type which indicates a visible DCW in analyzed signal according to our classification, in prehypertensive and hypertensive class it occurs with a probability of 39.63% in Dataset 1 and 37.10% in Dataset 2. The class with a low presence in both datasets is C type which does not have a visible trend against BP sorting criteria. The class with a high occurrence rate is NC type which represents a barely visible DCW in the analyzed signal. It appears to be more frequent in Dataset 1 than in Dataset 2 for the same BP ranges. At this point, A and N types represent promising key features to distinguish between normotensive patients and hypertensive ones.

By evaluating the ANC type against age sorting criteria, different trends can be observed especially for C type which lacks against BP assessment. Taking both Tables 5 and 6 into consideration for a large age spectrum, it can be observed that this type occurs frequently in children under 10 years old and again in people over 40. Thus, C type does not have a linear trend by the current sorting criteria. For A type, the occurrence happens in an older population with a slightly positive trend. Again, NC type is the predominant one among age spectrums in both datasets. The other type, N, appears only in young persons as it is highlighted in Table 5 within the 20–29 age group.

With these two sorting criteria, BP and age group, the summary of ANC type allocation among the population is presented. The scope is to avoid biases, like the belief that N type occurs only in a young population. However, as it was shown, this parameter is not dependent exclusively on the age since it is also distributed among various systolic blood pressure values.

### 2.4. ANC Test Enchanted by Machine Learning

The final step to predict the systolic blood pressure category based on the defined signal morphology classification is to enchant the ANC Test results with a powerful detection tool. In this way multiple machine learning (ML) architectures have been evaluated using classification learner tools from Matlab™ Software. The tools contain the following models: Decision Trees, Logistic Regression, Discriminant Analysis, Naive Bayes, Support Vector Machine, Nearest Neighbor Classifiers, and Ensemble Classifier. We tested every model to obtain the best metrics. Since the aim of the present study was to detect systolic hypertensive persons, to fulfil Level 1—BP prediction, two categories were defined:

- Class I: Recordings with associated systolic BP value from 80 to 129 mmHg;
- Class II: Recordings with associated systolic BP value over 130 mmHg.

Therefore, in Class I are clustered systolic hypotensive and normotensive datasets while in Class II are clustered prehypertensive and hypertension ones where the threshold value is 130 mmHg. Multiple clustering for each known BP category was avoided due to the low number of datasets and for uneven distribution as it was shown in the histogram back in Figure 2.

The input parameters for ML models are represented by the results of the ANC Test™ where the occurrence rate for each type and DtS parameter were computed automatically on PPG signal. The output is represented by the previous defined BP binary clusters. Therefore, given an input PPG waveform, the developed algorithm extracted ANC features and then fed them into the ML model. The last step should give the result if the given PPG signal comes from a normotensive or hypertensive patient.

Since in pre-evaluation section ANC types have a different distribution among systolic BP groups, the last human intervention is to set which feature combination will give the best predicted results. The assessment was done in two ways: checking individual feature standalone against defined BP clusters and checking the combination of features. These two assessments were done separately for Dataset 1 and Dataset 2 but also in the case by merging the two. With this approach, the influence of the recordings distribution was evaluated: Gaussian against uniform. The overview of the entire process is described in the below Figure 4:

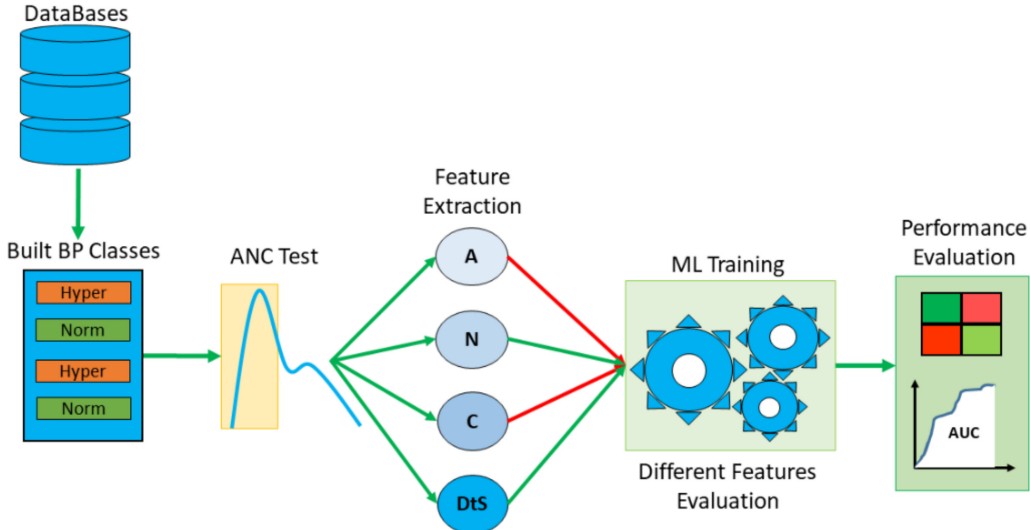

**Figure 4.** The procedure to evaluate features of ANC Test™ with ML models.

## 3. Results

### 3.1. Feature Standalone Evaluation

The first step to enchant the ANC Test is to feed the ML algorithm with standalone features. This will represent the starting point for the next assessment, representing at the same time, and the minimum performance can be obtained. The results are described in the Tables 7–9 for each dataset.

**Table 7.** ML metrics using ANC Test feature standalone for Dataset 1.

| Feature Selection | Model | Accuracy [%] | Precision [%] | Recall [%] | Specificity [%] | F1 Score | AUC |
|---|---|---|---|---|---|---|---|
| A | SVM—Gaussian | 67.1 | 65.6 | 60.6 | 71.6 | 0.63 | 0.64 |
| N | SVM—Gaussian | 56.6 | 88.4 | 48.8 | 76.6 | 0.63 | 0.57 |
| NC | SVM—Gaussian | 56.6 | 54.7 | 48.6 | 61.6 | 0.51 | 0.58 |
| C | SVM—Gaussian | 56.6 | 16.8 | 45.7 | 57.1 | 0.25 | 0.53 |
| DtS | KNN—Coarse | 58.4 | 45.3 | 51.8 | 61.8 | 0.48 | 0.55 |

**Table 8.** ML metrics using ANC Test feature standalone for Dataset 2.

| Feature Selection | Model | Accuracy [%] | Precision [%] | Recall [%] | Specificity [%] | F1 Score | AUC |
|---|---|---|---|---|---|---|---|
| A | SVM—Gaussian | 59.6 | 98.8 | 60.6 | 66.7 | 0.75 | 0.56 |
| N | SVM—Gaussian | 68.1 | 82.1 | 70.4 | 64.3 | 0.76 | 0.68 |
| NC | SVM—Linear | 63.3 | 89.3 | 64.7 | 62.5 | 0.75 | 0.64 |
| C | SVM—Gaussian | 59.6 | 97.6 | 60.3 | 50.0 | 0.75 | 0.5 |
| DtS | SVM—Gaussian | 60.3 | 84.5 | 60.7 | 43.5 | 0.71 | 0.56 |

**Table 9.** ML metric using ANC Test feature standalone for Merged dataset.

| Feature Selection | Model | Accuracy [%] | Precision [%] | Recall [%] | Specificity [%] | F1 Score | AUC |
|---|---|---|---|---|---|---|---|
| A | SVM Quadratic | 61.3 | 53.6 | 62.7 | 59.7 | 0.58 | 0.61 |
| N | SVM Cubic | 57.9 | 79.3 | 56.8 | 66.1 | 0.66 | 0.58 |
| NC | SVM Gaussian | 59.6 | 63.7 | 59.4 | 61.1 | 0.61 | 0.59 |
| C | Logistic Regression | 52.6 | 90.5 | 52.4 | 66.0 | 0.66 | 0.54 |
| DtS | Logistic Regression | 59.1 | 58.1 | 57.5 | 57.9 | 0.58 | 0.58 |

For Dataset 1, characterized by a Gaussian distribution of systolic BP values, A feature was the most accurate predictor in detecting hypertensive patients with an accuracy value of 67.1% using SVM Gaussian model. The F1 score and area under learning curve (AUC) was also the best factors in respect to the remaining parameters. The next ranked parameter is DtS which represent numerically the DCW strength in respect to the current SYS one. The obtained accuracy was 58.4% but with a weaker F1 score than the previous one. The remaining features obtained an accuracy of 56.6% but is closer to the accuracy of a random choosing with a 50% rate. Therefore, the last ones do not bring any standalone benefits.

For Dataset 2, built by a uniform BP value distribution, N feature was the most accurate predictor with a 68.1% rate also using SVM Gaussian model with an F1 score of 0.76. Additionally, it obtains the best AUC rate out of all parameters. Interestingly, the remaining features scored around 60% accuracy, also with a good F1 score. At this point, by comparing it, it can be concluded that a uniform distribution that feeds ML models gives a better performance than a Gaussian one.

The last evaluation of standalone feature was done by merging the two datasets. Since there were two different BP value distribution results, also in a Gaussian type, the accuracy results in special seem to be a mean between Dataset 1 and Dataset 2. The best accuracy was obtained by A type with a rate of 61.3% using an SVM Quadratic model. Although intuitively feeding an ML model with more samples is to increase its prediction rate, in the current context, the distribution of the recordings among desired output plays an important role in metrics too.

*3.2. Feature Combination Evaluation*

The last training set-up of ML models used the same approach with three dataset scenarios but with ANC feature combinations. The starting point was represented by the accuracy rank obtained in the standalone evaluation from Tables 7–9. The expectations of using combinations of two or more features is to increase the synergy for an ML model, thus the prediction rate should increase too. The results and metrics are shown in Tables 10–12.

**Table 10.** ML metrics using combination of ANC Test feature for Dataset 1.

| Features | Model | Accuracy [%] | Precision [%] | Recall [%] | Specificity [%] | F1 Score | AUC |
|---|---|---|---|---|---|---|---|
| A, DtS | Coarse Tree | 69.9 | 58.9 | 66.7 | 71.1 | 0.63 | 0.65 |
| N, DtS | Fine Tree | 58.4 | 45.3 | 54.4 | 62.9 | 0.49 | 0.56 |
| NC, DtS | Logistic Regression | 58.9 | 53.7 | 55.4 | 65.4 | 0.55 | 0.59 |
| C, DtS | Ensemble | 58 | 45.3 | 57.3 | 63.9 | 0.51 | 0.61 |
| A, N | Coarse Tree | 66.2 | 63.2 | 61.2 | 71.1 | 0.62 | 0.64 |
| A, N, DtS | Coarse Tree | 67.6 | 58.9 | 66.7 | 71.1 | 0.63 | 0.65 |
| A, NC | Coarse Tree | 68 | 63.2 | 63.2 | 71.8 | 0.63 | 0.68 |
| A, NC, DtS | Coarse Tree | 68.5 | 58.9 | 66.7 | 71.1 | 0.63 | 0.65 |
| A, N, NC | Coarse Tree | 67.6 | 61.1 | 63.0 | 70.9 | 0.62 | 0.67 |
| A, N, NC, DtS | Coarse Tree | 68 | 58.9 | 66.7 | 71.1 | 0.63 | 0.66 |
| A, N, NC, C, DtS | Coarse Tree | 68 | 58.9 | 66.7 | 71.1 | 0.63 | 0.66 |

**Table 11.** ML metrics using combinations ANC Test feature for Dataset 2.

| Features | Model | Accuracy [%] | Precision [%] | Recall [%] | Specificity [%] | F1 Score | AUC |
|---|---|---|---|---|---|---|---|
| A, DtS | Naive Bayes | 61.4 | 88.1 | 61.7 | 50.0 | 0.73 | 0.58 |
| N, DtS | Quadratic SVM | 67.1 | 83.3 | 73.7 | 68.9 | 0.78 | 0.69 |
| NC, DtS | Quadratic SVM | 63.4 | 77.4 | 64.4 | 51.3 | 0.70 | 0.62 |
| C, DtS | Gaussian SVM | 59.3 | 96.4 | 60.4 | 50.0 | 0.74 | 0.56 |
| A, N | Quadratic SVM | 70 | 82.1 | 69.7 | 65.9 | 0.75 | 0.66 |
| A, N, DtS | Quadratic SVM | 66.4 | 83.3 | 70.7 | 65.9 | 0.77 | 0.70 |
| A, NC | Cubic KNN | 62.9 | 73.8 | 67.4 | 54.2 | 0.70 | 0.61 |
| A, NC, DtS | Coarse Tree | 65 | 72.6 | 69.3 | 55.8 | 0.71 | 0.61 |
| A, N, NC | Quadratic SVM | 67.9 | 96.4 | 65.9 | 82.4 | 0.78 | 0.64 |
| A, N, NC, DtS | Quadratic SVM | 72.9 | 84.5 | 71.0 | 67.5 | 0.77 | 0.61 |
| A, N, NC, C, DtS | Quadratic SVM | 70.7 | 84.5 | 69.6 | 65.8 | 0.76 | 0.64 |

**Table 12.** ML metrics using combinations of ANC Test feature for Merged Dataset.

| Features | Model | Accuracy [%] | Precision [%] | Recall [%] | Specificity [%] | F1 Score | AUC |
|---|---|---|---|---|---|---|---|
| A, DtS | Ensemble | 62.4 | 54.2 | 61.4 | 59.2 | 0.58 | 0.61 |
| N, DtS | Gaussian SVM | 59.9 | 79.3 | 55.7 | 64.4 | 0.65 | 0.59 |
| NC, DtS | Gaussian SVM | 59.6 | 72.1 | 56.6 | 61.8 | 0.63 | 0.58 |
| C, DtS | Gaussian SVM | 57.9 | 56.4 | 60.1 | 59.2 | 0.58 | 0.63 |
| A, N | Weighted KNN | 59.1 | 81.0 | 56.2 | 66.3 | 0.66 | 0.60 |
| A, N, DtS | Gaussian SVM | 60.7 | 58.1 | 61.2 | 60.3 | 0.60 | 0.61 |
| A, NC | Quadratic SVM | 57.1 | 60.9 | 56.2 | 57.6 | 0.58 | 0.61 |
| A, NC, DtS | Ensemble | 61.8 | 58.1 | 59.4 | 59.2 | 0.59 | 0.60 |
| A, N, NC | Naive Bayes | 57.7 | 65.4 | 56.3 | 58.9 | 0.60 | 0.60 |
| A, N, NC, DtS | Ensemble | 63.2 | 59.2 | 61.3 | 60.8 | 0.60 | 0.62 |
| A, N, NC, C, DtS | Ensemble | 62.1 | 60.3 | 61.4 | 61.2 | 0.61 | 0.60 |

Using a Gaussian dataset, the best accuracy of the feature combinations, rising up to 69.9%, was the A, DtS case as shown in Table 10 and Figure 5a. The F1 score did not increase more than 0.63 in the standalone evaluation within Dataset 1;just the AUC metric did. The other feature combinations scored around 68% accuracy with the same F1 score. Increasing the number of features to feed the ML model does not give a boost in performance since many selections represent a redundant role.

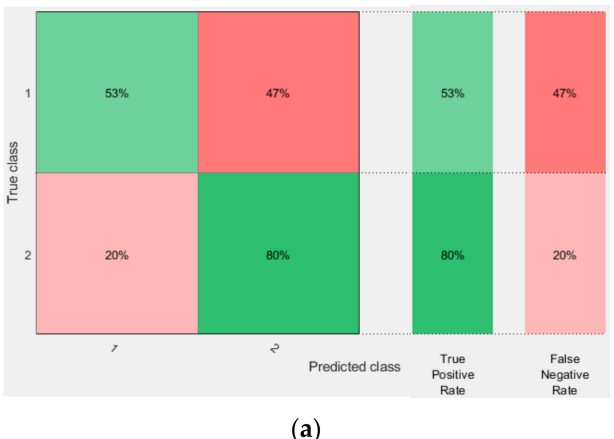

(**a**)

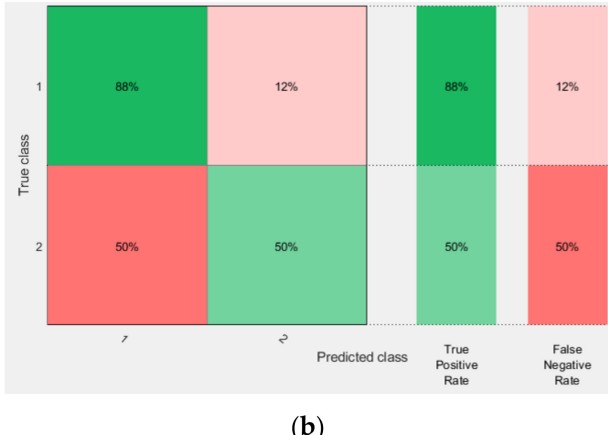

(**b**)

**Figure 5.** Best accuracy obtained in Dataset 1—Coarse Tree (**a**) and Dataset 2—Quadratic SVM (**b**).

Moving to Dataset 2, the best accuracy obtained of 72.9% comes from using 4/5 ANC parameters as shown in Table 11 and Figure 5b. Closer to a 70% accuracy score also came with 2/5 features: A and N type. The F1 score is very close to standalone evaluation and the AUC metric. Overall, the accuracy increases in the best case only by about a 4.8% rate

rather than in Dataset 1, of 27%. As in the first evaluation, certain combinations of ANC features have a redundant role. In the context of wearable devices, it is desired to reduce the computation power, therefore, the best predictors of hypertensive would be A and N type. On the other hand, where computation power does not represent a constrain like a server computation, the selection with best performance would be implemented: A, N, NC, and DtS parameters.

For the merged strategy evaluation, the overall accuracy scored between the performance of Dataset 1 and Dataset 2. The best accuracy was 63.2% similar to in the Dataset 2 case by A, N, NC, and DtS parameters with an F1 score of 0.60. In a nutshell, the prediction of detecting prehypertensive or hypertensive patients is around 70% even if the different datasets have been used with individual measurement protocols. The designed algorithm by sorting various signal morphologies in defined classes prove a good performance for BP prediction of Level 1. Therefore, the ANC Test™ represents a good starting point in this health metric evaluation.

## 4. Limitations

The first limitation is given by the measurement protocol used in individual datasets. The first key parameter in this topic is given by the PPG sensor LED wavelength since the light beam targets different depths of tissue. It is known that green light is able to reach 1–2 mm immediately after the epidermis layer and the infra-red (IR) light is the strongest one which can reach the deepest layers of dermis while the red wavelength is situated in the middle in terms of performance. Therefore, the PPG obtained with IR light contained the most hemodynamic information which can be obtained with this technique. The second electrical related parameter is the sampling frequency used by the processing chain. Dataset 1 used 1 kHz where a high-detail level of signal was obtained, while Dataset 2 used 125 Hz. By lowering Fs, the gradient read by FD would be affected, thus the ANC results would suffer too. The third critical parameter which also has been tested, is the cut-off frequency for filtering the PPG signal apart from noise. By lowering this frequency, the details of the PPG are again attenuated, leading to a different ANC results value.

The second limitation is given by the number of samples and its distribution. Even if around 70% accuracy has been obtained with a number of 219 and 140 patients, respectively, it is known that more data will strength the ML model by giving a better performance. The type of samples distribution plays an advantageous role as was shown by comparing Gaussian Dataset 1 and uniform Dataset 2. Although it is intuitive to merge two or more datasets in order to increase the number of samples, it is not a winning strategy. With a different measurement protocol and a different recordings distribution it will weaken or limit the overall accuracy of the ML model.

## 5. Conclusions

The Level 1 of blood pressure prediction using a PPG signal has been demonstrated by using a signal morphology classification strategy. Designing the ANC Test™ represents a good tool in this assessment, where the selection criteria were given by the anacrotic type and dicrotic wave behaviour in the current cardiac cycle. By enchanting the results of the designed test with ML models, the best performance of systolic BP prediction was 72.9% with room for more improvement. Sorting signals morphology into clearly defined classes not only represents a good numerical tool for data processing, but also shows a snapshot of signal characteristic distributions across a desired group analysis.

The primary aim of this study was to integrate the obtained results into the digitalization trend within a health monitoring area. Thus, a standalone PPG signal gives good insight into the cardiovascular state if the right investigation method is applied. An agreed protocol should be adopted for the signal processing chain in order to evaluate health metrics such as systolic blood pressure. In this way, the minimum amount of cardiovascular information will be set by using optical sensors. As proof, using multiple datasets with

different measurement protocols does not represent an interim solution to increase the accuracy of the ML model.

Having at its base simple sorting rules, the ANC Test™ can be implemented into wearable devices like smartwatches and smartbands in order to detect the hypertension phase. For best performance, the test can run on the servers to move the processing power away from tiny devices. During 24 h, a user experiences various blood pressure values triggered by different physical activities, emotional states, or even by drugs. Therefore, the benefit of the designed test is to act like a blood pressure alarm for users in need. From a health monitoring level point of view, this cardiovascular marker can track during a whole day in different scenarios, rather than a simple visit to a physicist's office or by using classical devices like oscillometric blood pressure measurements. Another use of the presented study is to extract other key metrics of the signal morphology which are correlated with a certain cardiovascular disorder apart from hypertension. We aim to improve this test by searching for other key features which can be used in further evaluations, but to also raise the test's accuracy closer to medical grade requirements.

**Author Contributions:** Conceptualization, L.E., D.D., S.S. and L.D.; methodology, L.E. and S.S.; software, L.E.; validation, L.E., L.D., S.S. and S.H.; formal analysis, L.E. and S.S.; investigation, L.E.; resources L.E.; data curation, L.E.; writing—original draft preparation, L.E.; writing—review and editing, L.E. and L.D.; visualization. L.E., D.D. and L.D. supervision, L.D. and D.D.; project administration, L.D.; funding acquisition, L.D. All authors have read and agreed to the published version of the manuscript.

**Funding:** This paper publication is funded by University Politehnica of Bucharest; PUBART project.

**Institutional Review Board Statement:** Not applicable.

**Informed Consent Statement:** Not applicable.

**Data Availability Statement:** Not applicable.

**Conflicts of Interest:** The authors declare no conflict of interest.

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
