# Peer review of "Hypertension Detection Based on Photoplethysmography Signal Morphology and Machine Learning Techniques"

_applsci, doi:10.3390/app12168380_

Round 1

Reviewer 1 Report

The concept of the work and the presented results are clear to me.

  Sometimes irritating can be the lack of explanation of the abbreviations used, eg DCW etc, which fortunately appear in the following paragraphs. Please pay attention to this.

I would suggest adding a fragment of how the data obtained by you can translate into specific technological solutions (watches, wristbands). Please specify in what medical situations it could be applied and how it would improve the health of the societies.

I encourage you to continue research on the accuracy of blood pressure testing in everyday devices.

Reviewer 2 Report

The paper deals with a hypertension detection based on photoplethysmography. The database have 359 individuals with a 70% of accuracy using the PPG signal and learning machines techniques.

The best results is obtained using Quadartic SVM whit 72.9% of accuracy.

The paper is well presented with representative results.

Author Response

Dear Reviewer,

Thank you for the comments and positive feedback regarding our paper entitled “Hypertension Detection Based on Photoplethysmography Signal Morphology and Machine Learning Techniques”. Those comments are all valuable and very helpful for revising and improving our paper, as well as the important guiding significance to our research. Now, we are submitting the revised version after seriously considering the comments of the Reviewers. We hope that this version will improve the quality of our manuscript and make it more acceptable for publication.

Sincerely,

The Corresponding Author

Reviewer 3 Report

Occasionally, writing is awkward and hard to understand. I suggest consulting with a native English speaker to improve the quality of writing. For example, line #52-61 needs rewriting and maybe even further clarification.

Please add references for: ‘vasodilator and vasomotor drugs changing the shape of PPG signal’

Line #71, define DCW

It seems that the article specifically addresses the systolic blood pressure. Please specify.

Data bases used in this study are from stationary patients, mostly in ICU and rarely moving. Given the sensitivity of PPG signal to movement, the accuracies reported in this article would not be necessarily applicable to general population and for the day-to-day kind of activities. Please discuss.

Authors should add a few lines describing the relationship between different stages of a PPG wave and different stages of cardiac cycle. This info should also be included in Figure 3

Not sure what line #150-2 is trying to say. Same with line #171-172

Regarding the equation on line #155:

Description not clear

What does gradient mean in this context? Is it the slope?

Is gsys the average of FD for the anacrotic phase?

what is gDCW? Is it similarly the average value for FD for a specific interval? if yes, how is that interval defined for N type and for NC type?

Did the author investigate a possible correlation between the A and N features, as these characteristics might be correlated since they are both related to aging -- the former appears with aging and the latter diminishes with aging?

Was the age at all considered in the model? If not, how are tables 5 and 6 relevant?

In line #210-5, was a program used to extract features of each PPG wave or was it done by visual inspection? Please clarify.

Confusion matrix and accuracy are not sufficient metrics for the evaluation of the model performance. Authors should at least include F1 score, sensitivity and specificity and the area under ROC curve.

Authors should additionally report Cohen’s kappa as the number of normotensive vs hypertensive patients are significantly different.

Furthermore, it would be relevant for some of these metrics to be reported separately for the normotensive and hypertensive patients.

Reviewer 4 Report

The authors claimed a hypertension diagnosis tool based on PPG via machine learning. I would like to recommend a resubmission.

The manuscript is not well formatted/prepared:

(1) There are many errors "[Error! Reference source not found.]. ".

(2) Table 1 is not named properly.

(3) Where is ref 8 cited?

(4) In Figure 3, subfigures are not named properly.

The issues below need to be addressed.

Line 93: Why the four databases were selected?

Line 126-132: Any reference to support the classification of A, N, C, N/C?

Line 199: "multiple Machine Learning (ML) architecture has been evaluated using Classification Learner from Matlab Toolbox"

What ML architectures were used and why? Any figures to show the accuracy?

Round 2

Reviewer 4 Report

As my comments are properly addressed, I recommend acceptance of the manuscript.